# Effect of Arginine Vasopressin on Human Neutrophil Function Under Physiological and Sepsis-Associated Conditions

**DOI:** 10.3390/ijms26062512

**Published:** 2025-03-11

**Authors:** Sophie-Marie Haile, Michael Gruber, Gabriele Bollwein, Benedikt Trabold

**Affiliations:** 1Department of Anaesthesiology, University Hospital Regensburg, 93042 Regensburg, Germany; sophie-marie.haile@ukr.de (S.-M.H.); michael.gruber@ukr.de (M.G.);; 2Department of Internal Medicine II, University Hospital Regensburg, 93042 Regensburg, Germany

**Keywords:** neutrophils, sepsis, arginine vasopressin, chlorobutanol, NETosis, ROS, migration, live cell imaging, antigen expression (CD11b, CD62L)

## Abstract

This study examines how different concentrations of arginine vasopressin (AVP) and its preservative chlorobutanol (ClB) impact the immune functions of human polymorphonuclear neutrophils (PMNs), which are crucial in the immune response, particularly in sepsis. Using a model to simulate the physiological, sepsis-related, and therapeutic AVP levels in plasma, we analysed how AVP and ClB affect PMN activities, including reactive oxygen species (ROS) production, NETosis, antigen expression, and migration. PMNs were isolated from whole human blood and assessed using flow cytometry and live cell imaging. The results indicated that neither AVP nor ClB significantly affected PMN viability, antigen expression, NETosis, or ROS production in response to N-Formylmethionine-leucyl-phenylalanine, or fMLP, and tumour necrosis factor alpha. In the migration assays, concentration-dependent effects were observed. At physiological AVP levels, PMN migration showed no reduction, while the sepsis-associated AVP levels initially reduced migration before returning to the baseline or even increasing. The therapeutic AVP concentrations showed similar migration to that in the controls, while high concentrations progressively inhibited migration. ClB, regardless of its concentration, enhanced PMN migration. These findings suggest that AVP during sepsis may impair PMN migration, potentially contributing to tissue damage and systemic complications. This highlights AVP’s role as a possible immune modulator in complex immune responses.

## 1. Introduction

In recent decades, modern medical research has yielded significant insights into the factors influencing the human immunological defence system. One particularly interesting area of research is the effect of endocrine signals on neutrophil granulocytes (PMNs). These cells constitute the majority of the white blood cells and play a key role in the immune response to infection, as well as in the pathogenesis of sepsis. Given that arginine vasopressin (AVP) is both an endogenous hormone and a potential therapeutic option in sepsis, it may play a pivotal role in the pathophysiology of diseases such as sepsis.

PMNs are the physiological first-line defence of the human immune system. As part of the non-specific immune system, they possess a multitude of defence mechanisms. These mechanisms include migration towards the site of infection, the production of oxygen radicals (ROS) that directly eliminate pathogens, and the release of neutrophil extracellular traps (NETosis), which facilitate the capture of pathogens. These mechanisms are effective in eliminating pathogens. The correct extent of the induced PMN defence mechanisms is essential: if excessive and uncontrolled, they can cause significant damage to the surrounding tissue, resulting in host damage. These mechanisms of PMNs therefore act as a double-edged sword: uncontrolled immune defence mechanisms have the potential to trigger severe symptoms such as organ failure that would not be caused by the pathogen itself. These phenomena are evident in the pathogenesis of sepsis patients. Conversely, some PMN functions may be suppressed, potentially impeding the ability to effectively combat pathogens [1].

Sepsis represents a significant health concern, characterised by a critical dysregulation of the immune system in response to an infection, often triggered by a bacterial infection. In sepsis, the host’s immune system overreacts to the infection, leading to systemic inflammation, immune cell activation, and, in some cases, tissue damage. A potential complication of sepsis is organ failure, particularly that of the lungs and heart. In response to infection, pro-inflammatory cytokines, such as tumour necrosis factor-alpha (TNF-α) and interleukins, are released, triggering further immune responses. This dysregulated immune response can lead to a paradoxical combination of both excessive inflammation and immune suppression. In cases of severe sepsis, cardiovascular stabilising therapies such as AVP administration are employed to elevate blood pressure and provide cardiac support [2]. The precise impact of AVP on PMNs in a septic context remains to be elucidated. In order to enhance and refine the existing therapeutic modalities, it is imperative to examine the impact of AVP on PMNs in the context of septic conditions.

Arginine vasopressin is a neurohormone produced in the hypothalamus of the brain and released into the bloodstream at varying concentrations, contingent on the physiological or pathological condition [3,4,5]. The existing literature provides evidence that AVP has functions beyond those of an antidiuretic and vasoconstrictor. It may also exert a direct effect on leukocytes [6].

It is established that AVP can be bound by leukocyte surface proteins, thereby triggering effects at the molecular and cellular levels in leukocytes [6,7,8,9,10]. As evidenced in the literature, AVP primarily binds to mononuclear cells; however, the binding of AVP to PMNs has also been described [11,12]. The AVP-binding sites of leukocytes have been the subject of analysis in a number of studies. For example, the AVP-binding site of splenic lymphocytes has been observed to exhibit similarities to the known AVP receptor V1 [13]. The AVP-binding sites of CD4-positive T-cells have already been identified [10]. Additionally, it has been demonstrated that AVP functions as a chemoattractant for monocytes [14] and exerts a stimulatory effect on natural killer cells [9]. It has been demonstrated that rats with deficiencies in the AVP hormone exhibit a general increase in PMNs in their blood during ontogenesis, while the number of lymphocytes decreases [10]. In braindead rats, the cell surface expression (CD11b) on the PMNs of AVP-treated rats exhibited lesser elevation than that in the control group [15]. It can thus be postulated that AVP exerts an inhibitory effect on PMNs, although further detailed investigations are required to substantiate this hypothesis. Only a limited number of studies have addressed the direct impact of AVP on the defensive mechanisms of PMNs. To date, modifications of the Boyden chamber, an often-used model of migration research, have been employed. This approach allows for the quantification of the respective endpoint of migration in a single migration direction [16,17].

To the best of our knowledge, no comprehensive scientific study has yet been conducted to investigate the effect of different AVP concentrations (in the physiological state, during sepsis, during therapeutic AVP administration) on various neutrophil functionalities (chemotaxis, ROS production, myeloperoxidase (MPO) release, and NETosis). This knowledge is essential to gain a comprehensive understanding of the intricate interplay between defective PMN functionality and the potential impact of AVP on it, thereby enabling the development of novel therapeutic approaches. Furthermore, it is imperative to comprehensively investigate the effect of the intensive care drug AVP on PMN functionality to facilitate the future adaptation of sepsis therapy. Therefore, we conducted an in vitro investigation with healthy donors. In addition to flow cytometry, the method of live cell imaging was selected due to its ability to examine not only the migration of PMNs over time but also the temporal sequence of the various defence strategies employed by these cells. The concentrations of AVP employed were meticulously selected to optimally represent disparate AVP plasma concentration states, encompassing a healthy physiological state, the pathological state of sepsis, and the state under continuous AVP-infusion administration. Furthermore, the AVP concentrations employed in this study were aligned with those previously documented in the literature to ensure its comparability [3,5,18,19,20,21,22,23,24,25,26,27,28,29,30].

Hence, the present experimental study investigates whether AVP exerts a potential immunomodulatory effect on PMNs at physiological, pathological (septical), therapeutic (under continuous AVP-infusion administration during sepsis therapy), and supraclinical concentrations and the sole potential effect of chlorobutanol, a common component of the drug AVP, on PMNs.

## 2. Results

### 2.1. Flow Cytometry

#### 2.1.1. The Percentage of Propidium Jodide (pj)-Stained Cells

The proportion of pj-stained cells remained unaltered in the presence of either ClB or AVP/ClB, irrespective of the respective concentration (n = 5–10 per group). Therefore, no discernible differences were observed between the individual concentrations of the substances. The proportion of all pj-stained PMNs was found to be below 9% on average. 

#### 2.1.2. Antigen Expression

No effect was observed on the expression of the antigens CD11b and CD62L of the substances ClB and AVP/ClB at any of the concentrations analysed (n = 7–10 each). 

#### 2.1.3. Reactive Oxygen Species Production

Neither ClB nor AVP/ClB showed a significant concentration-dependent effect on ROS formation. For the sake of clarity, all of the concentrations of the respective substance groups were summarised and considered as one group (see Figure 1). ROS production could not be triggered directly by either ClB or AVP/ClB without a stimulation substance (n = 6–28 per group). The ROS production triggered by N-Formylmethionine-leucyl-phenylalanine and tumour necrosis factor alpha (fMLP+TNFα) demonstrated no discernible difference between the individual drug groups. The phorbol 12-myristate 13-acetate (PMA)-induced ROS production of the PMNs in the control group (n = 12) differed significantly from that in those treated with ClB (n = 35; in pairwise comparison; *p* = 0.017), as well as the PMNs treated with AVP/ClB (n = 35; in pairwise comparison; *p* = 0.02), by 55% (ClB) and 64% (AVP/ClB), respectively. No significant difference was observed in the ROS production between the PMNs treated with ClB and those treated with AVP/ClB.

### 2.2. Live Cell Imaging

#### 2.2.1. Quantification of ROS Production and NETosis

T_max_ROS was identified in all groups prior to the parameter ET_50_MPO and before ET_50_NETosis. The individual ClB concentrations (cf. Table 1) demonstrated no statistically significant differences in the parameters T_max_ROS, ET_50_NETosis, or ET_50_MPO. Consequently, the respective ClB concentrations were aggregated into a single group. The incubation of the PMNs with ClB resulted in a significantly (*p* < 0.01) delayed ET_50_MPO (delta (control vs. ClB study group) ET_50_MPO = −58.17 min) compared to that in the control group. The incubation with ClB did not result in any significant alterations in the T_max_ROS and ET_50_NETosis parameters when compared to those in the control group. The PMNs of the AVP/ClB 1.24 pg/mL group exhibited a tendency to display a similar behavioural pattern with regard to the parameters T_max_ROS, ET_50_NETosis, and ET_50_MPO to that observed in the ClB test groups. The PMNs of the AVP/ClB 12.4 pg/mL group exhibited a tendency towards premature T_max_ROS in comparison to that in the control group (delta (control vs. AVP test group) T_max_ROS = −71.19 min). However, the other two parameters remained unchanged in comparison to those in the control group. The two highest AVP concentrations also exhibited an earlier T_max_ROS than that in the control group. The number of n in each group is indicated in Table A1, while the mean values are presented in Table 1.

**Table 1 ijms-26-02512-t001:** Mean values in [min] of the parameters T_max_ROS, ET_50_MPO, and ET_50_NETosis; n for each condition can be found in Table A1.

Experimental Group	T_max_ROS	SD	ET_50_MPO	SD	ET_50_NETosis	SD
control	107.22	65.51	165.44	47.56	250.23	49.71
ClB 1.24	119.80	25.40	219.14	14.86	214.26	39.80
ClB 12,400	93.66	38.74	225.06	25.00	227.21	16.75
Total ClB	100.55	36.88	223.61	32.23	230.80	127.40
AVP/CLB 1.24	129.37	43.90	276.77	60.03	260.86	32.75
AVP/CLB 12.4	36.03	40.66	139.70	1.24	209.84	78.54
AVP/CLB 124	109.55	48.71	161.62	43.15	229.43	75.72
AVP/CLB 1240	135.29	61.22	165.90	23.38	323.51	132.08
AVP/CLB 12,400	61.16	29.46	185.95	45.83	317.59	140.30
AVP/CLB Max	94.39	10.03	171.62	24.68	218.27	17.37

#### 2.2.2. Migration

The migration parameters track length, straightness, track displacement of X, and track displacement of Y exhibited a decline with the passage of time in all of the analysed groups (cf. Figure 2, Figure 3, Figure 4 and Figure 5). The significant *p*-values of all of the migration parameters in all investigated groups in comparison to those in the control group over the entire observation period are listed in Table 2 and are explained in more detail below.

(a)Track displacement of Y

The parameter track displacement of Y demonstrated no notable discrepancy in comparison to that in the control group, exhibiting no greater deviation than their cell diameter (15 µm) across all of the groups analysed. This was observed over the entire observation period and at all time intervals (cf. Figure 2).

(b)Track length
i.ClB compared to the control, total observation timeA comparison of the track lengths over the entire observation period revealed a significant difference (*p* < 0.001 in each case) between the experimental groups with ClB offset PMNs and the PMNs in the control group (Table 2).ii.ClB compared to the control, individual periodsThe results demonstrate a statistically significant increase (*p* < 0.001) in the track length for each concentration of ClB, with the greatest change occurring during the first two half-hour intervals (cf. Figure 3).iii.AVP/ClB compared to the control, total observation timeA comparison of the track length over the entire observation period revealed a significant difference (*p* < 0.001) between all of the AVP/ClB test groups, with the exception of those spiked with 12.4 pg/mL AVP/ClB, and the PMNs of the control group (Table 2).iv.AVP/ClB compared to the control, individual periodsFurthermore, the track length was markedly increased by 1.24 pg/mL AVP/ClB during the initial three half-hour periods (cf. Figure 3). As the AVP/ClB concentration increased, the track length exhibited a progressive decline, resulting in some PMNs spiked with higher AVP/ClB concentrations having significantly shorter track lengths than those of the control (cf. Table 2). The PMNs that were spiked with a concentration of 12.4 pg/mL AVP/ClB exhibited a significantly (*p* < 0.01) reduced track length compared to that in the control group during the initial 30 min. During the observation period of 31 to 60 min, the track lengths of the experimental group were found to be similar to those of the control group. During the observation periods of 61–90 min and 91–120 min, the track lengths were found to be significantly (*p* < 0.01) higher.v.AVP/ClB compared to ClB, total observation timeA direct comparison of the AVP/ClB group with the corresponding ClB group revealed no significant difference in the track length between the PMNs of the AVP/ClB 1.24 group and the PMNs of the corresponding ClB experimental group (ClB 1.24 group) over the entire observation period. The PMNs of the AVP/ClB 12,400 group exhibited a markedly reduced track length in comparison to that in the PMNs of the corresponding ClB group (ClB 12,400) throughout the observation period, with a statistically significant difference (*p* < 0.001).vi.AVP/ClB compared to ClB, individual periodsA direct comparison of the AVP/ClB group with the corresponding ClB group revealed no significant difference in the track length between the PMNs of the AVP/ClB 1.24 group and the PMNs of the corresponding ClB experimental group (ClB 1.24) at any time point. The PMNs of the AVP/ClB 12,400 group exhibited a significantly (*p* < 0.001) reduced track length in comparison to the PMNs of the corresponding ClB group (ClB 12,400) throughout the entire observation period.


(c)Straightness
i.ClB compared to control, total observation timeThe PMNs of all ClB test groups exhibited a statistically significant difference (*p* < 0.001 in each case) in straightness compared to that in the control group over the entire observation period.ii.ClB compared to the control, individual periodsThe PMNs of the 1.24 ClB and 12,400 ClB group exhibited a markedly higher degree of straightness (*p* < 0.001 in each case) compared to that in the control group during the initial three half-hour intervals.
iii.AVP/ClB compared to the control, total observation timeThe PMNs of all of the AVP/ClB test groups exhibited a statistically significant difference (*p* < 0.001 in each case) in straightness compared to that in the control group over the entire observation period.iv.AVP/ClB compared to the control, individual periodsIn the first three half hours, the PMNs incubated with 12.4 pg/mL AVP/ClB or higher exhibited a significantly reduced straightness (cf. Figure 4) in comparison to that in the PMNs in the control group. The incubation of the PMNs with 1.24 pg/mL AVP/ClB resulted in a significantly higher straightness at all time intervals (*p* < 0.01 in each case) compared to that in the PMNs in the control group. The straightness of the cells demonstrated a decline with an increase in the AVP/ClB concentration. The greatest reduction in straightness was observed in the PMNs incubated with the highest AVP/ClB concentration (AVP/ClB Max).v.AVP/ClB compared to ClB, total observation timeThe PMNs of the AVP/ClB 12,400 group exhibited a markedly diminished straightness throughout the observation period, as evidenced by a *p*-value of less than 0.01 in each instance, in comparison to that in the PMNs of the corresponding ClB group (ClB 12,400 group).vi.AVP/ClB compared to ClB, individual periodsA direct comparison demonstrated that the PMNs incubated with 1.24 pg/mL AVP/ClB exhibited markedly elevated straightness in the second (*p* < 0.05) and third half-hour (*p* < 0.01) periods in comparison to the PMNs incubated with the corresponding ClB concentration (ClB = 1.24). The cells spiked with 12,400 pg/mL AVP/ClB demonstrated this significantly (*p* < 0.01 in each case) in the first three half-hour periods when compared to the PMNs incubated with the corresponding ClB concentration (ClB = 12,400).


(d)Track displacement of X
i.ClB compared to the control, total observation timeThroughout the observation period, all of the PMNs spiked with ClB showed a significantly higher track displacement of X than that of the PMNs in the control group (cf. Table 2).ii.ClB compared to the control, individual periodsAll of the PMNs that were exposed to pure ClB exhibited a positive track displacement of X at all time points, indicating a trajectory towards the reservoir with the chemoattractant. A disparity of greater than one PMN cell diameter from the control group was observed during the initial two half-hour intervals. As the time progressed, the values of the track displacement of X parameter converged. From the fourth half hour (from 91 min after the commencement of microscopy), the track displacement of X parameter of the ClB test groups no longer exhibited a statistically significant difference from that of the control group, nor did it demonstrate a difference of less than one PMN diameter. In the initial half hour, all of the PMNs spiked with ClB exhibited a markedly elevated track displacement of X in comparison to that in the control group (*p* < 0.01) (cf. Figure 5).iii.AVP/ClB compared to the control, total observation timeThe PMNs spiked with AVP/ClB showed a significantly different track displacement of X than that of the PMNs in the control group at the lowest and the three highest concentrations used throughout the observation period (cf. Table 2). The PMNs incubated with 12.4 or 124 pg/mL AVP/ClB demonstrated no statistically significant difference in the track displacement of X parameter when compared to that in the control group over the entire observation period.iv.AVP/ClB compared to the control, individual periodsWith the exception of the PMNs belonging to the highest-AVP-concentration group, all of the PMNs that had been spiked with AVP/ClB demonstrated a positive track displacement of X at all time points, indicating a movement towards the reservoir with the chemoattractant. The PMNs incubated with the highest AVP concentration exhibited a negative track displacement, corresponding to a distance of less than one PMN cell diameter. A difference of greater than one PMN cell diameter from the control group was observed during the initial two half-hour periods. As the time progressed, the values of the track displacement of X parameter for all groups converged. From the fourth half hour (91 min after the commencement of microscopy), the track displacement of X parameter of the AVP/ClB experimental groups no longer exhibited a statistically significant difference from that of the control group, nor did it show a difference of less than one PMN diameter.In the initial 30 min, the PMNs treated with 1.24 pg/mL AVP/ClB exhibited a markedly elevated track displacement of X in comparison to that of the control group (*p* < 0.01 each). Higher AVP/ClB concentrations demonstrated a similar or significantly reduced track displacement of X in comparison to that in the control group (cf. Figure 5).v.AVP/ClB compared to ClB, total observation timeThe PMNs of the AVP/ClB 1.24 group demonstrated no statistically significant difference when compared to the PMNs of the corresponding ClB group (ClB 1.24) over the entire observation period.vi.AVP/ClB compared to ClB, individual periodsIn the initial 30-minute period, the PMNs of the AVP/ClB 1.24 group exhibited a markedly reduced track displacement of X in comparison to that in the corresponding ClB group (ClB 1.24). Subsequently, no notable discrepancy was observed. In the initial four half-hour intervals, the PMNs of the AVP/ClB 12,400 group exhibited a markedly diminished track displacement of X in comparison to that in the corresponding ClB group (ClB 12,400) (*p* < 0.01 in the first two half-hour intervals, *p* < 0.05 in the third and fourth half-hour intervals).



ijms-26-02512-t002_Table 2Table 2The *p*-values for the migration parameters (track length, straightness, and track displacement of X) of ClB- or AVP/ClBClB-incubated PMNs compared to those of the control group over the entire observation period. N for each condition can be found in Table A2.Experimental GroupTrack LengthStraightnessTrack Displacement of XClB 1.24<0.001<0.001<0.001ClB 12,400<0.001<0.001<0.001AVP/ClB 1.24<0.001<0.001<0.001AVP/ClB 12.4No significance<0.001No significanceAVP/ClB 124<0.001<0.001No significanceAVP/ClB 1240<0.001<0.001<0.05AVP/ClB 12,400<0.001<0.001<0.001AVP/ClB Max<0.001<0.001<0.001


## 3. Discussion

The present study investigated the effect of physiological, pathological, and pharmacological arginine vasopressin (AVP) concentrations on the defence mechanisms (ROS, NETosis, antigen presentation, and migration) of PMNs. This knowledge may contribute to a better understanding of the dynamics of diseases and symptoms triggered by excessive or reduced neutrophil granulocyte activity. Therefore, it could help to understand the effects of AVP administration in the intensive care of patients with complex diseases such as sepsis and thus help to develop or adapt therapeutic options. It is known that first-line drugs such as catecholamines have an inhibitory effect on PMN defence strategies such as ROS production and PMN migration [31,32,33,34,35]. It is therefore suggested that the subsequent susceptibility of patients to infection is increased by the administration of catecholamines. The effect of AVP, the second-line therapy, on PMN defence mechanisms has not been adequately investigated. Furthermore, the impact of chlorobutanol, a prevalent component (0.5%) of commercially available AVP, was examined, given the existing literature indicating that ClB can modulate molecular cell mechanisms [36,37].

### 3.1. The Proportion of Dead Cells, Antigen Production, ROS Production, and NETosis in Relation to AVP/ClBClB and ClB

The findings of this study demonstrate that neither ClB nor AVP/ClBClB, irrespective of their concentration, exert any influence on the proportion of dead PMNs. Furthermore, the findings of this study demonstrate that the antigen expression (CD11b, CD62L) of non-activated PMNs, which is crucial for diapedesis, cannot be regulated by either ClB or AVP/ClB. To the best of our knowledge, there is no existing literature that has analysed the effect of AVP or ClB on PMN antigen expression (CD11b, CD62L) or on PMN diapedesis.

Moreover, the findings of this study indicate that there is no impact of ClB or AVP/ClB on the quality of ROS production in the absence of a stimulus, with fMLP or with fMLP and TNFα. These observations are in accordance with the findings of [16]. Conversely, the present study demonstrates that both ClB and AVP/ClB, irrespective of their concentration, are capable of enhancing the ROS production triggered by PMA. ROS production can be triggered by a variety of substances and via different molecular metabolic pathways. In contrast to the PMA-activated signalling pathway as a direct activator of protein kinase C (PKC), the fMLP-activated downstream pathway is not a linear molecular pathway [38,39,40,41]. Rather, the individual molecules formed in the process influence each other and lead to increased ROS production via different molecular pathways. Additionally, TNF-α has been demonstrated to augment the production of ROS by PMNs [42]. Given that ClB did not affect the fMLP- or fMLP& TNFα-induced ROS production in our experimental setting, irrespective of its concentration, it is probable that ClB only interferes with the molecular mechanism of PKC-activated ROS production and not with the entire fMLP downstream pathway. To the best of our knowledge, no research to date has investigated the effect of ClB on ROS production by PMNs. The available evidence suggests that ClB does not interfere with the molecular calcium balance of the cells, as indicated by studies investigating its effect on calcium homeostasis in muscle cells [43,44]. As the ROS production activated by fMLP, which is triggered, among other things, by the intracellular release of calcium, was not influenced by ClB in our study, it can be concluded that ClB does not influence the calcium balance of other cells and that, at least partially, it does not affect it in PMNs.

The present study demonstrates that chlorobutanol exerts an effect whereby protein kinase C or its downstream pathway is likely to be involved in ROS production. Given the multiplicity of the molecular mechanisms that trigger ROS production in vivo and the fact that the ROS production triggered by fMLP is the most closely aligned with this in vivo phenomenon, it can be posited that the in vivo effect of ClB on PKC or its downstream pathway is not a significant factor in ROS production. Given that AVP/ClB also demonstrated no effect beyond that of ClB, it can be posited that AVP exerts no considerable influence on the ROS production of PMNs in vivo.

The results of the present study are also consistent with the findings of the previous literature, which indicates that ROS production always occurs prior to MPO release, and that this occurs in close proximity to DNA release [45,46,47]. The present study demonstrates that this sequence is not susceptible to alteration by either ClB or AVP/ClB. In the present study, a delay in the MPO release of the PMNs incubated with ClB was found to be statistically significant in comparison to that in the control group. This does not indicate that the process of NETosis was delayed by ClB, as the ET_50_NETosis parameter was comparable to that of the control. Nevertheless, the data suggest that the intervals between PMN defence mechanisms may be variable and potentially susceptible to alteration by the addition of a substance. It is noteworthy that the PMNs incubated with a sepsis concentration exhibited a tendency towards premature ROS production. The premature production of ROS may result in tissue damage and consequently the progression of sepsis. To the best of our knowledge, there is no existing literature that has investigated the temporal aspect of PMN defence mechanisms.

### 3.2. Chemokinesis and Chemotaxis in Relation to AVP/ClB and ClB

The migration of PMNs is an essential component of effective pathogen control. In the literature, a distinction is made between chemokinesis (the description of undirected movement or a change in migration in the presence of chemicals) and chemotaxis (directed movement along a CA gradient) [48]. As the migration process is well defined by these two technical terms, the results are categorised accordingly. Table 3 presents the assignment of the migration parameters employed in this study (track displacement of X, track displacement of Y, track length, and straightness) to the two migration terms.

**Table 3 ijms-26-02512-t003:** Classification of the migration parameters chemotaxis and chemokinesis.

Chemotaxis	Chemokinesis
track displacement of X, track displacement of Y, straightness	track displacement of Y, track length, straightness

A comparison of the migration results presented in this study with those reported in the previous literature reveals a general alignment with the findings of studies that employed comparable conditions [16,17]. In their study, Doherty et al. demonstrated that 0.5 U/mL AVP (equivalent to 500 pg/mL AVP) had no impact on a 60 min PMN migration pathway in the absence of a chemoattractant gradient. A comparison of the aforementioned result with the present one reveals that it is most closely aligned with the parameter track displacement of Y (no gradient of a chemoattractant) observed in the AVP/ClB 124 group after 60 min: no difference in chemokinesis was observed in comparison to that in the control group. The results of [16] are also consistent with the present work in many respects. The mean PMN migration width after 30 min with the existing fMLP gradients (comparable to the track displacement of X parameter) of all experimental groups incubated with AVP is approximately the same (with a difference of less than one PMN diameter) as the mean migration data for comparable groups in the present study (ref. [16] mean: 69.4 µm; in the present study: 54.61 µm).

In the described literature [16,17] on the effect of AVP on PMN migration, the methodology of a Boyden chamber was used. This is inferior to the experimental model of the present study (live cell imaging). In a Boyden chamber, only the endpoint of a migration pathway in a single direction can be visualised, which means that much of the information about the actual migration pathway is lost. This means that important dynamic aspects of cell movement and interactions with the environment cannot be captured. In contrast, the migration model used in our work is much closer to the in vivo situation. It not only allows cell migration to be observed in real time but also provides comprehensive information on several parameters simultaneously. These include migration itself, ROS production, MPO production, and NETosis. These parallel measurements allow for a more precise and detailed investigation of the migration properties and defence mechanisms of the cells. Nevertheless, these are all in vitro experiments and should therefore be interpreted with caution when applied to in vivo situations. Moreover, the tested PMNs were obtained from healthy subjects rather than, for example, sepsis patients. Nevertheless, the results could be crucial for gaining important insights.

The present study demonstrates that chemokinesis and chemotaxis of PMNs are influenced by both AVP and ClB. These effects were most pronounced at the outset of the observation period. This was likely due to the fact that both chemokinesis and chemotaxis decreased independently of the experimental groups with increasing time in the present setting. Consequently, the effects of AVP and ClB could be determined primarily at the beginning of the observation period. This phenomenon has also been described in previous publications on in vitro migration studies [45,47]. The effect of AVP on PMN migration appears to be concentration-dependent, while the effect of ClB is concentration-independent (cf. Table 4):

The findings of this study demonstrate that ClB, irrespective of its concentration, enhances the chemokinesis and chemotaxis of PMNs. This study indicates that the physiological AVP concentration also increases the chemokinesis of PMNs. This was corroborated by the chemokinesis of the PMNs in the AVP/ClB 1.24 group, which was increased over the entire observation period. The PMNs of the corresponding ClB 1.24 group exhibited this phenomenon only at the outset of the observation period. It can thus be posited that a physiological AVP concentration exerts a sole increasing effect on the migration of PMNs, independent of ClB. With regard to chemotaxis, this study demonstrates that the physiological AVP concentration has no inhibitory effect on chemotaxis over the entire observation period. It is crucial to differentiate between the two migration terms, chemotaxis and chemokinesis, in this context. The PMNs of groups 1.24 ClB and 1.24 AVP/ClB showed a similar effect in terms of chemotaxis, so one could wrongly assume that the effect on migration was due to ClB alone. However, chemokinesis indicates that the physiological AVP concentration does have a sole enhancing effect on PMN migration.

This study also demonstrates that high concentrations of AVP neutralise or inhibit the activated effect on PMN migration. Higher AVP/ClB concentrations exhibited a decline in both chemotaxis and chemokinesis with increasing concentration in comparison to these values in the PMNs incubated with pure ClB. The corresponding data on AVP/ClB and ClB in comparison revealed that the impact of AVP at high concentrations (AVP/ClB 12,400) surpassed that activated effect, resulting in a reduction in migration ability. Consequently, the presence of ClB resulted in an increase in both chemotaxis and chemokinesis, whereas the administration of AVP at higher doses led to a reduction in these processes. The findings of this study demonstrated that elevated AVP concentrations not only negated the impact of ClB but also impeded the migration of PMNs in comparison to that in the control group. In comparison to the control group, the PMNs incubated with supraclinical AVP concentrations demonstrated a notable reduction in both chemokinesis and chemotaxis. The migration of the PMNs was found to be reduced in the presence of supraclinical AVP concentrations. Once PMNs are no longer able to migrate to the site of a pathogen, they can no longer effectively eliminate it. This could also result in host damage.

**Table 4 ijms-26-02512-t004:** Qualitative overview of the impact of AVP/ClB or ClB incubation at various concentrations on the chemokinesis and chemotaxis of PMNs in comparison to those in the control group (cf. Figure 3, Figure 4 and Figure 5).

**Clinical Significance of the Analysed AVP-Concentration**	**Chemokinesis**	**Chemotaxis**	**Experimental Group**
physiological AVP-plasma concentration	increased	not decreased	AVP/ClB 1.24
AVP-plasma concentration during Sepsis	dependent on time		AVP/ClB 12.4
AVP-plasma concentration during continuous AVP-infusion	no effect	no effect	AVP/ClB 124
Supraclinical AVP- concentration	decreased	decreased	AVP/ClB 1240
Supraclinical AVP- concentration	decreased	decreased	AVP/ClB 12,400
Supraclinical AVP- concentration	decreased	decreased	AVP/ClB Max
	Chemokinesis	Chemotaxis	Experimental group
Chlorobutanol concentration	increased	increased	ClB 1.24
	increased	increased	ClB 12,400

The PMNs incubated with AVP concentrations corresponding to the AVP plasma concentrations under continuous AVP-infusion administration (AVP/ClB 124) demonstrated comparable chemotaxis and chemokinesis to these properties in the PMNs of the control group. In our setting, the effect of ClB and AVP on the migration of the PMNs was almost cancelled out under AVP-Perf AVP plasma concentrations. The stable state between the inhibitory effect of a high AVP concentration and the activating effect of AVP and ClB meant that they cancelled each other out. These results are of interest, as ClB is often present in AVP-infusions, and therefore the clinical situation under sepsis therapy was simulated by our model.

It is notable that the migration of the PMNs incubated with the AVP plasma concentration reflective of sepsis conditions (AVP/ClB 12.4) was not in line with the expectations “higher AVP concentration results in lower migration”. Given that the other AVP/ClB test groups demonstrated a decline in chemokinesis and chemotaxis with an increasing concentration, it was anticipated that the chemokinesis and chemotaxis of the PMNs incubated with the AVP plasma concentration of sepsis (AVP/ClB 12.4) would fall between these values for the physiological AVP plasma concentration (AVP/ClB 1.24) and the AVP plasma concentration during continuous AVP-infusion administration (AVP/ClB 124). However, the PMNs of the sepsis group (AVP/ClB 12.4) exhibited chemotaxis and chemokinesis at the outset of the investigation that was markedly diminished compared to what was expected. The results therefore indicate that the AVP concentration, which corresponds to the phase of sepsis, may initially reduce the migration potential of PMNs. A reduction in the migration capacity of PMNs during sepsis has also been documented in the literature [1].

Directed migration is a prerequisite for the effective defence of immune cells. If this decreases or if the PMNs move in a completely undirected manner, they can no longer deploy their defence capabilities in a targeted manner. This could result in an inability to effectively combat pathogens at the site of inflammation. This could lead to a surge of pathogens, or alternatively, the unguided defence mechanisms of the PMNs could inflict damage upon the body’s own tissue. Both of these can lead to the development of sepsis. It is therefore unsurprising that the PMNs with an AVP concentration corresponding to the situation during sepsis, in which the function of the PMNs is altered according to the literature [1], demonstrated restricted migration. An AVP concentration which corresponds to sepsis may act as a trigger for this inhibition of PMN function. It is noteworthy that a tenfold higher AVP concentration (under continuous AVP-infusion) did not result in an initial decline in PMN migration. If the hypothesis that reduced migration contributes to the development of sepsis is valid, early AVP administration may potentially prevent the initial reduction in migration and thereby facilitate more effective pathogen control. This would lend support to the theory that the early administration of AVP at the onset of sepsis may prove beneficial to patients [49].

In conclusion, our study shows that AVP has no effect on the viability of PMNs or their defence mechanisms, including the production of adhesion molecules (CD11b/CD62L) and fMLP- and fMLP&TNFα-induced ROS production. Furthermore, our study demonstrates that AVP at different concentrations significantly affects the migratory properties of PMNs. Notably, while our experimental AVP concentration series showed a dose-dependent decrease in PMN migration, the sepsis-associated AVP levels led to a significantly stronger migration impairment than that expected from our in vitro data. In contrast, the AVP plasma concentration observed during AVP perfusion therapy did not result in such a pronounced reduction in PMN migration. These results highlight the complex role of AVP in immune regulation and suggest that this knowledge may contribute to the development of effective therapeutic applications for inflammatory diseases such as sepsis.

## 4. Materials and Methods

**Granulocyte preparation and incubation.** All of the experiments were approved by the Ethics Committee of the University of Regensburg, Germany (reference number 06/169). Prior to the commencement of this study, the 21 healthy volunteers, who donated approximately 5.5 mL of venous blood, were informed about the potential risks associated with the collection of blood samples. The volunteers were seated during the blood collection process. At least 30 min of rest was observed prior to the collection of blood, during which time no sporting activity was undertaken. The blood was collected into a Li-Heparin S monovette (Sarstedt AG & Co., Nümbrecht, Germany) and subsequently purified at room temperature via density gradient centrifugation at 756 G, utilising the Lympho PBMC Spin Medium and Leuko Spin Medium from pluriSelect Life Science (Leipzig, Germany). Subsequently, the PMN concentration was determined using a Neubauer chamber and recorded in RPMI 1640 medium (PAN-Biotech GmbH, Aidenbach, Germany) with 10% foetal calf serum (FCS; PAN-Biotech GmbH, Aidenbach, Germany), resulting in a cell concentration of 18 × 10⁶ cells/mL.

In order to achieve the desired drug concentrations in the experimental preparations, dilution series of the respective drugs were prepared in advance using phosphate-buffered saline (PBS, Sigma, Darmstadt, Germany) and subsequently added to the experimental preparations. The AVP (Amomed Pharma GmbH, Vienna, Austria) concentrations illustrated in Figure 6 were selected to represent physiological, pathological, and supraclinical AVP concentrations, as well as the AVP plasma concentration during continuous AVP-infusion administration. A plasma concentration of 1.24 pg/mL AVP is representative of the physiological level. A plasma concentration of 12.4 pg/mL AVP is indicative of a patient in the early stages of sepsis. A plasma concentration of 124 pg/mL AVP is indicative of continuous AVP-infusion administration. All of the other AVP concentrations analysed correspond to supraclinical concentrations and were emulated for the purpose of comparison with the existing literature. Given that AVP always contained 0.5% ClB as a solvent, the corresponding test groups are henceforth designated as “AVP/ClB”. The concentrations of ClB (0.063, 0.63, 6.32, 63.2, and 632 pg/mL) are in proportion to the 0.5% ClB (Sigma, Steinheim, Germany) content present in 1.24, 12.4, 124, 1240, and 12,400 AVP/ClB. The maximum concentration, denoted as Max, represented the highest level of the analyte that could be achieved in a given test batch. For the method of live cell imaging (migration, T_max_ROS, ET_50_NETosis, and ET_50_MPO), this corresponded to an AVP concentration of 1,030,000 pg/mL and a ClB concentration of 54,000 pg/mL. In the FACS experiments (pj, CD11b, CD62L, and ROS production), this equated to an AVP concentration of 621,000 pg/mL and a ClB concentration of 32,600 pg/mL.

**Flow cytometry**. For the ROS and viability measurements, 10 µL of the previously obtained PMNs was added to 0.5 mL of PBS, and an additional 5 µL of PBS was added for the control group. Alternatively, 5 µL of each of the drug dilution series was added to obtain the concentrations described in Figure 6. Subsequently, the cell suspension was incubated for 10 min on a tilting roller at 15 rpm and centrifuged, and the resulting solution was mixed with 100 µL of PBS. Subsequently, 5 µL of SNARF (Sigma Aldrich Chemie GmbH, Steinheim, Germany), 5 µL of dihydrorhodamine 123 (DHR 123, Molecular Probes, Inc., Eugene, OR, USA), and 5 µL of pj (Molecular Probes, Inc.) were added. The cell suspension was then incubated for 10 min on a tilting roller at 15 rpm, centrifuged, and taken up in 100 µL of PBS. Then, 5 µL of the stimulus substance, either fMLP (Sigma Aldrich, St. Louis, MO, USA), fMLP and TNFα (1 µg/mL, Pepro Tech Inc., Rocky Hill, KY, USA), PMA (10 µM, Sigma Aldrich), or PBS as the control, was added.

To quantify the cell surface antigens CD11b and CD62L, 10 µL of the PMN suspension, 500 µL of PBS, and an additional 5 µL of PBS (control) or 5 µL of the drug dilution series were combined to achieve the concentrations depicted in Figure 6. The suspension was incubated for 10 min on a tilting roller at 15 rpm and then centrifuged, and the supernatant was removed. The pellet was resuspended in 100 µL of PBS. Subsequently, 2.5 µL of PBS or 2.5 µL of monoclonal antibodies (labelled with either phycoerythrin (PE, DC11b ICRF44, BioLegend, San Diego, CA, USA) or fluorescein isothiocyanate (FITC, CD62L DREG-56, BioLegend)) was added.

The ROS production, viability, and antigen measurements were conducted in duplicate using the FACSCalibur flow cytometer from BD Biosciences (Erembodegem, Belgium) and the CellQuest Pro software, version 5.2 (BD Biosciences, San Jose, CA, USA). The analysis was conducted using the FlowJo software (version 10.0.7, LLC, Ashland, OR, USA), Microsoft Excel (version 2023), and the IBM SPSS Statistics software (Statistical Package for the Social Sciences, IBM Corporation, New York, NY, USA, version 25).

**Live cell imaging**. For the migration experiments and the detection of ROS production and NETosis, the PMNs were spiked with 1 μM dihydrorhodamine 123 DHR 123 (ROS production) and 0.5 μg/mL of an APC-conjugated anti-human MPO antibody (REA491, Miltenyi Biotec, Auburn, CA, USA). A solution of 5 μg/mL 40.60-diamidino-2-phenylindole (DAPI, Sigma-Aldrich) was added to a total volume of 60 µL 10× MEM (Sigma, Steinheim, Germany), 60 µL H_2_O, 30 µL NaHCO_3_, 150 µL RPMI with 10% FCS, and 150 µL collagen I gel (1.5 mg/mL PureCol, Advanced BioMatrix Inc., Carlsbad, CA, USA). Furthermore, 5 µL of PBS (as the control) or 5 µL of a drug dilution series was added to achieve ClB concentrations of 0.063 pg/mL and 632 pg/mL, as well as the AVP/ClB concentrations as described in Figure 6.

Subsequently, 6.5 µL of the aforementioned preparation was added to the channel of an IBIDI 3D µ-Slide migration chamber. The IBIDI 3D µ-Slide comprises three migration chambers, thereby enabling the simultaneous execution of three distinct experiments. The reservoir on the right was consistently filled with 65 µL of the RPMI 1640 cell medium with 10% FCS. Furthermore, the left reservoir was supplemented with the chemoattractant fMLP (10 nM), thereby ensuring a linear chemoattractant concentration gradient in the gel in the X direction. The same chemoattractant concentration was therefore maintained in the Y direction. Subsequently, the migration of the PMNs was observed automatically for a period of five hours using a Leica DMi8 microscope (Leica Microsystems, Wetzlar, Germany), a motorised microscope stage, a Leica DFV9000 camera, and the Leica Application Suite X software platform, version 3.4.2.18368. To this end, three fluorescence images and one phase contrast image were captured automatically at 30 to 40 s intervals. A climate chamber from IBIDI was employed to maintain a constant temperature of 37 °C and a CO_2_ concentration of 5%. The mean time elapsed between the initiation of drug contact and the commencement of microscope observation was 52.0 min (standard deviation: 8.02; minimum: 41 min; maximum: 71 min).

**Image data analysis**. The data generated during live cell imaging were analysed using Imaris version 9.0.2 software (Bitplane, Zurich, Switzerland). The software employed a semi-automatic recognition and tracking process for the migration paths of the PMNs in the resulting phase contrast series images, subsequently calculating the following parameters:

Track length in µm (the length of the migration route of each individual cell):$$Track\;length=\sum_{t=t_{F}+1}^{t_{L}}\sqrt{{Dx(t,t-1)}^{2}+{Dy(t,t-1)}^{2}}$$

Track displacement of X in µm (the distance between the start and end point of the analysed cell on the *x*-axis):$$D_{x}\left(t_{1},t_{2}\right)=P_{X}\left(t_{1}\right)-P_{X}\left(t_{2}\right)$$

Track displacement of Y in µm (the distance between the start and end point of the analysed cell on the *y*-axis):$$D_{y}\left(t_{1},t_{2}\right)=P_{y}\left(t_{1}\right)-P_{y}\left(t_{2}\right)$$

Straightness (the fraction of Euclid track length and total track length showing the cell’s tendency to migrate directly; higher factors refer to straighter lines), without a unit, in a range from 0 to 1:$$straightness=\frac{track\;displacement}{track\;length}$$

D = track displacement, $T_{L}\;$= last time index of the track, $T_{F}\;$= first time index of the track, $P_{X}\left(t\right)\;$= the x-position of an object at time index *t*, and $P_{y}\left(t\right)\;$= the y-position of an object at time index *t*.

The PMNs were observed and analysed over the entire observation period and in half-hour blocks (0–30 min, 31–60 min, etc.). Cell migrations below the diameter of a PMN (10–15 µm) were considered to be random movements or artefacts and thus were not included in the migration data. Subsequent analyses were conducted using Excel and SPSS. To eliminate potential artefacts, the 50 cells that exhibited the greatest displacement were selected, and a threshold of a track duration greater than 1000 s and a minimum track length of 25 micrometres was applied. Experiments in which Imaris identified fewer than 25 cells or there were incomplete recordings were excluded from the subsequent analysis. As in the study published in [47], the PMN functions were quantified in Excel using the size of the area of the detected fluorescence of the recorded fluorescence images as a function of time. The parabolic curve observed in graphical quantification of ROS production allows for the quantification of ROS production according to T_max_. T_max_ROS represents the time in minutes at which the maximum ROS production of the PMNs occurs. The detected fluorescence areas corresponding to NETosis (DAPI staining) and myeloperoxidase (MPO staining) exhibited a sigmoidal curve, and thus these parameters were quantified according to the respective ET_50_ value (ET_50_NETosis for DAPI and ET_50_MPO for MPO). The ET_50_ value indicates the time at which half of the maximum detectable fluorescence area is reached. Incomplete datasets were excluded from the analysis.

**Statistical analysis.** The number of measurements (n) for each experiment is provided in the tables in the appendix (Table A1 and Table A2) or described in the Results section. The descriptive and inductive statistics, as well as all of the graphs, were generated with the assistance of the statistical software package SPSS. The standard deviation (SD) is provided in parentheses alongside the mean values. The figures illustrate the mean values and the corresponding 95% confidence intervals for the respective groups under analysis. Lilliefors tests were conducted to ascertain the presence of a normal distribution. The Levene test was employed for the purpose of calculating the variance homogeneity. In the event of a normal distribution, parametric tests (*t*-tests), a one-way analysis of variance with Bonferroni correction (for given variance equality), or Dunnett-T3 correction (for variance inequality) were calculated for the respective control group. In the event of the lack of a normal distribution, non-parametric tests were employed (the Mann–Whitney-U test or the Kruskal–Wallis test with pairwise comparison). Unless otherwise stated, significant differences are indicated by a *p*-value of less than 0.05. In the figures, the symbol “*” indicates a statistically significant difference (0.001 < *p* < 0.05) in comparison to the respective control group. Meanwhile, the symbol “§” indicates a statistically significant difference of *p* < 0.001. The error bars in the graph represent the 95% confidence intervals.

## Figures and Tables

**Figure 1 ijms-26-02512-f001:**
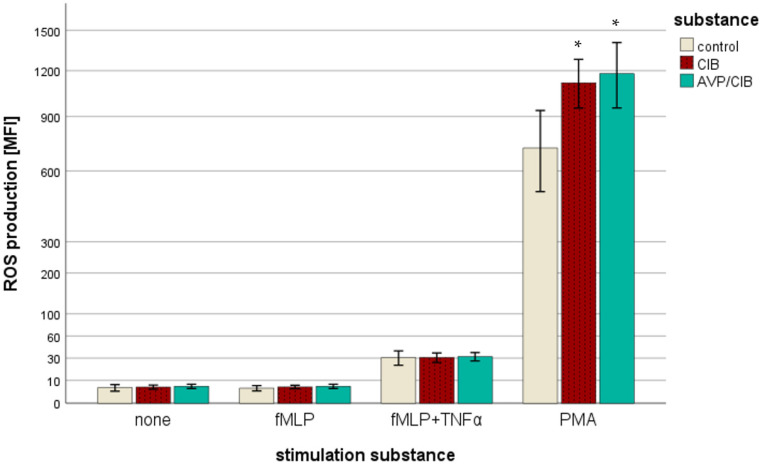
Reactive oxygen species (ROS) production in [MFI] (mean fluorescence intensity). “none” corresponds to ROS production without the activation of cells: without the addition of fMLP (N-Formylmethionine-leucyl-phenylalanine), TNFα (tumour necrosis factor alpha), or PMA (phorbol 12-myristate 13-acetate). The control group is neutrophil granulocytes (PMNs) without the addition of chlorobutanol (ClB) or arginine vasopressin and its preservative chlorobutanol (AVP/ClB). The drug groups comprise all concentrations under investigation. The number of subjects included in the study ranged from 10 to 35. The y-axis is scaled according to a power transformation with an exponent of 0.5. The symbol “*” indicates a statistically significant difference within the range 0.001 < *p* < 0.05.

**Figure 2 ijms-26-02512-f002:**
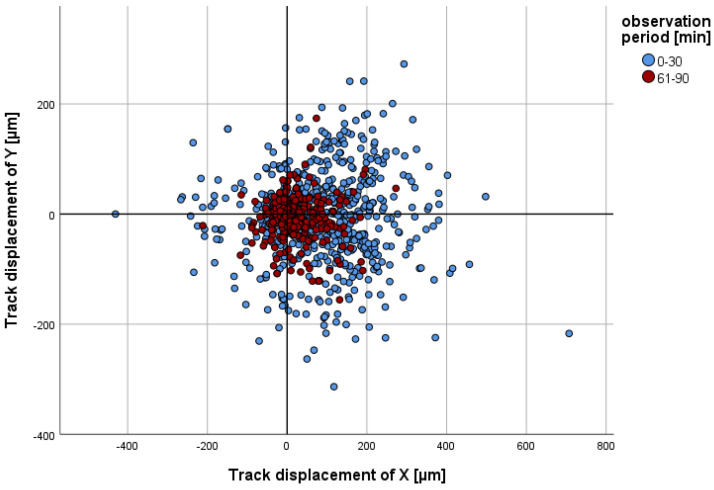
The x/y coordinates of individual PMNs in the control group at minute 30 (blue, n = 786) or minute 90 (red, n = 725) are presented in comparison to their starting point (x = 0 and y = 0) at minute 0 or minute 61. A positive track displacement of X is indicative of movement towards the chemoattractant.

**Figure 3 ijms-26-02512-f003:**
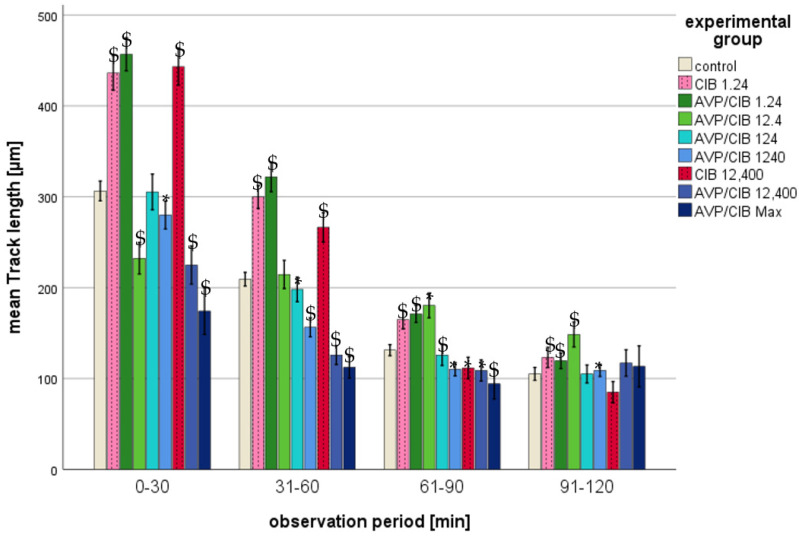
Mean track length in micrometres of ClB or AVP/ClB (various concentrations)-incubated PMNs in comparison to PMNs in the control group over time. The symbol “*” indicates a statistically significant difference within the range 0.001 < *p* < 0.05. The symbol “$” indicates a statistically significant difference with *p* < 0.001 in comparison to the respective control group.

**Figure 4 ijms-26-02512-f004:**
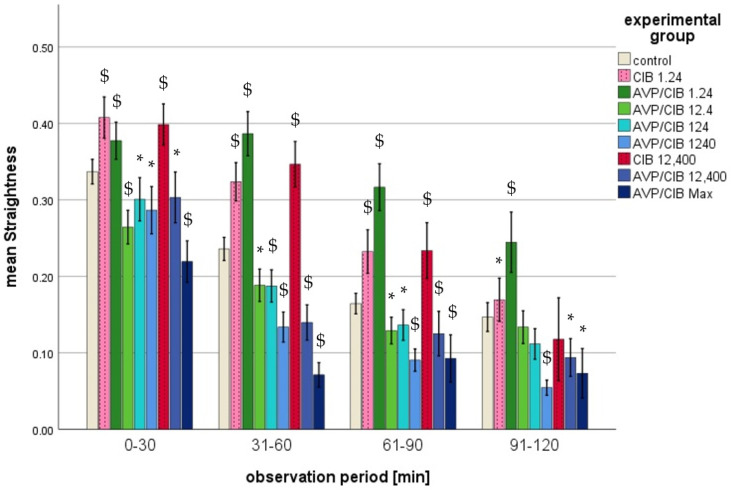
Mean straightness (range: 0 to 1) of ClB or AVP/ClB (various concentrations)-incubated PMNs in comparison to that in the PMNs of the control group over time. The symbol “*” indicates a statistically significant difference within the range 0.001 < *p* < 0.05. The symbol “$” indicates a statistically significant difference with *p* < 0.001 in comparison to the respective control group.

**Figure 5 ijms-26-02512-f005:**
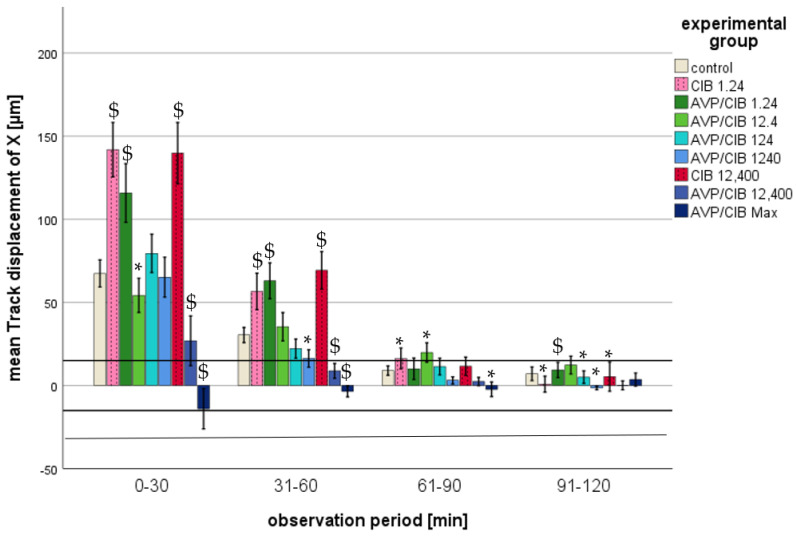
Mean track displacement of X in µm of ClB- or AVP/ClB (various concentrations)-incubated PMNs in comparison to that in the PMNs of the control group over time. The symbol “*” indicates a statistically significant difference within the range 0.001 < *p* < 0.05. The symbol “$” indicates a statistically significant difference with *p* < 0.001 in comparison to the respective control group.

**Figure 6 ijms-26-02512-f006:**
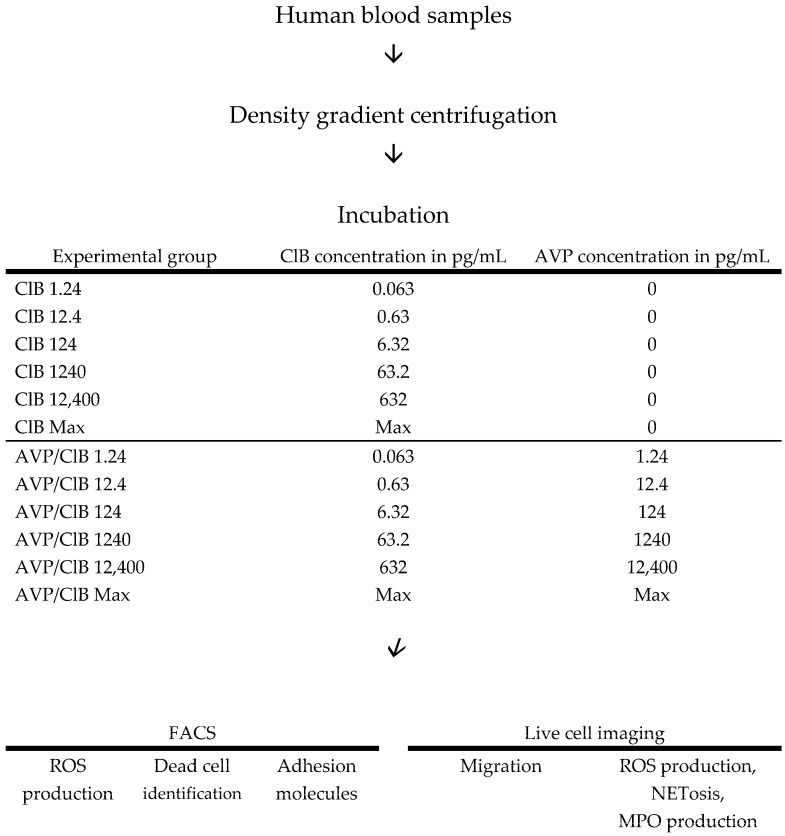
Flowchart illustrates the examination techniques employed and an overview of the drug concentrations utilised in the respective test groups. “Max” corresponds to the maximum concentration that could be achieved in each experiment.

## Data Availability

The data presented in this work are available on request from the corresponding author.

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
