# Peer review of "Effect of Arginine Vasopressin on Human Neutrophil Function Under Physiological and Sepsis-Associated Conditions"

_ijms, 2025, doi:10.3390/ijms26062512_

Round 1
Reviewer 1 Report
Comments and Suggestions for Authors
This manuscript reports studies on the effects of vasopressin (V) and preservative chrorobutanol (C) on polymorphonuclear neutrophil (PMN) functions, an important area of research in critical care medicine and sepsis in particular. The investigators draw the conclusion that V alters PMN motility functions but not phenotype, survival, or PMN activation properties. Based on the data presented on Figure 2, and 4-6, and Table 4, however, it is unclear how V, which is used in the V/C combination, has any effects that are different than the effects of C. On the other hand, C appears to be the active compound depending on the concentration used, the importance of which in the drug's formulation and mechanism of actions of the commonly used pharmaceutical is not directly addressed.
1. The contention in the manuscript that various concentrations of V and/or C combinations in vitro simulate or recapitulate physiological or septic conditions is not warranted. The study was performed on PMNs from healthy individuals.
2. A V alone group to discern the effects of V from those of C alone is not included. Studies are with C alone or V/C combination
3. Some experiments (eg Figure 2, Table 1 and 4) summarize data or present pooled data from different concentrations of V or C tested without clear statistical justification of why this is valid. The V/C 12.4 for example in Table 1 shows lower values for ROS and MPO and fluctuation at other combination compared to, I assume untreated, control. It is not discernible whether these fluctuations represent technical inconsistency or biological variation. Similar concerns with fluctuating data shown in Figure 4-6.
3. The data referred to in 3.2.1 and 3.2.2 is not shown.
4. Data in Figure 1 suggests that C is prooxidant in PMA-stimulated neutrophils. Why were data pooled? What is the minimal concentration of the compounds that has an effect? What is the concentration of stimulants in these experiments? What is the effect of V and C in other PKC-mediated functions of PMNs?
5. The Methods indicate that PMNs were isolated from 21 donors, yet the n reported for the different varies from 6-28 and 35 in different experiments. In most experiments p values are reported but not the n number.
6. Experimental details and reagents utilized to measure and data output expression as TmaxROS, ET50NETosis, or ET50MP) are not described in methods.
7. The C1.24 group in Figures 4-6 is compared to V/C 1.24 and C12400, but not at intermediate concentration. It is unclear from this data that V has any activity different than C.
8. Discussion is too lengthy and conclusions in section 3.1, for example, is a literature review rather than findings focused.
Author Response
Please accept our apologies, as an incorrect file was recently uploaded (currently listed on the website as "author-coverletter-42713208.v1.pdf"). Since we are unable to delete it, we kindly ask you to disregard this version and refer to the updated file instead. We apologize for any inconvenience this may have caused.

Reviewer 2 Report
Comments and Suggestions for Authors
Dear authors!
I congratulate you on your work and encourage you to continue your research. There is a need for deep intervention on the written material, I made some recommendations, respectively requests in the following.
Summary
It must bring into discussion notions that are known until the moment of this research. Abbreviating Vasopressin with V leaves room for confusion, and it is not common to use it in studies, the abbreviation for Vasopressin is AVP (arginine vasopressin).
Also Chlorbutanol cannot be abbreviated with C, either we use the name as such Chlorbutanol, or we find another abbreviation such as ClB.
Use abbreviations not explained in the text. such as NETosis or fMLP and TNFα. These must be explained at the first writing, in order for the reader to understand exactly what you are referring to.
The summary must be clear, to arouse the reader's interest, using abstract abbreviations and unclear sentences, we will lose his curiosity.
The summary must be completely rewritten.
Introduction
The introduction is clearer, but with many notions that should not be highlighted and others that are superficially treated.
About the physiopathological response of sepsis and the mechanisms involved in the host's response to infection, should be described more clearly.
The role of arginine vasopressin and the receptors it acts on should be explained in the introduction.
Figure 1 should be moved to the Method chapter.
Lines 91-94 are unclear, i.e. it must be defined what sepsis, severe sepsis, septic shock means??
Also an unprecedented term of Therapeutical Sepsis ??
Material and Method
In the Material and Method chapter, we must describe very clearly the method used, the patients included in the study, the inclusion and exclusion criteria. Likewise, there are many elements that distract attention, the line of study must be simple and clear, leaving no room for interpretations
What does PBS mean? Line 532-533
The type of study must also be specified, because the description is not clear and the reader must deduce which study it is.
What should be clear from the method is which patients did you take over? healthy, with sepsis, severe sepsis, septic shock?
There are elements that are treated extensively and other important ones that are not touched!
The material part and the simplified method, because there are many details that make us lose the essential.
Laboratory methods should not be described exhaustively, because we lose the essence of the study.
The presented tables are not sufficiently explained.
The discussions are mixed with the method and the results. I think it is necessary to deeply review this part of the study as well.
The conclusions of the study are not addressed, which must refer strictly to your research. You should also address the limitations of this study.
Dear authors!
I congratulate you on your work and encourage you to continue your research. There is a need for deep intervention on the written material, I made some recommendations, respectively requests in the following.
Summary
It must bring into discussion notions that are known until the moment of this research. Abbreviating Vasopressin with V leaves room for confusion, and it is not common to use it in studies, the abbreviation for Vasopressin is AVP (arginine vasopressin).
Also Chlorbutanol cannot be abbreviated with C, either we use the name as such Chlorbutanol, or we find another abbreviation such as ClB.
Use abbreviations not explained in the text. such as NETosis or fMLP and TNFα. These must be explained at the first writing, in order for the reader to understand exactly what you are referring to.
The summary must be clear, to arouse the reader's interest, using abstract abbreviations and unclear sentences, we will lose his curiosity.
The summary must be completely rewritten.
Introduction
The introduction is clearer, but with many notions that should not be highlighted and others that are superficially treated.
About the physiopathological response of sepsis and the mechanisms involved in the host's response to infection, should be described more clearly.
The role of arginine vasopressin and the receptors it acts on should be explained in the introduction.
Figure 1 should be moved to the Method chapter.
Lines 91-94 are unclear, i.e. it must be defined what sepsis, severe sepsis, septic shock means??
Also an unprecedented term of Therapeutical Sepsis ??
Material and Method
In the Material and Method chapter, we must describe very clearly the method used, the patients included in the study, the inclusion and exclusion criteria. Likewise, there are many elements that distract attention, the line of study must be simple and clear, leaving no room for interpretations
What does PBS mean? Line 532-533
The type of study must also be specified, because the description is not clear and the reader must deduce which study it is.
What should be clear from the method is which patients did you take over? healthy, with sepsis, severe sepsis, septic shock?
There are elements that are treated extensively and other important ones that are not touched!
The material part and the simplified method, because there are many details that make us lose the essential.
Laboratory methods should not be described exhaustively, because we lose the essence of the study.
The presented tables are not sufficiently explained.
The discussions are mixed with the method and the results. I think it is necessary to deeply review this part of the study as well.
The conclusions of the study are not addressed, which must refer strictly to your research. You should also address the limitations of this study.
Reviewer 3 Report
Comments and Suggestions for Authors
The aim of this manuscript is to investigate the effect of Vasopressin on PMN (polymorphonuclear leukocyte) migration. Blood samples were collected from 21 healthy individuals, and PMNs were isolated and incubated with varying concentrations of Vasopressin or its preservative. The findings indicate that physiological concentrations of Vasopressin stimulate PMN migration, whereas pathological levels or therapeutic doses suppress PMN migration. While this study presents interesting results, the following suggestions should be addressed:
- What is the sample size ("n") for each group in the cell migration assay?
- Considering that all PMNs were isolated from 21 healthy individuals, how were individual samples identified? Were the samples pooled or analyzed separately?
- Additional methods for ROS measurement should be performed to confirm the findings.
- A sepsis-mimic model, such as LPS-induced inflammation, should be tested on PMNs. Without such experiments, the current findings only demonstrate the effects of Vasopressin on PMN migration under general conditions and not under sepsis-associated conditions.
Round 2
Reviewer 2 Report
Comments and Suggestions for Authors
Dear authors!
First of all, congratulations for your exceptional work. I reread the article and the changes made with pleasure. I think that the article is sufficiently clear and well explained.
In table 4, the first column, I think V is AVP?, according to the other abbreviations and should be changed.
The conclusions should be highlighted in a separate chapter.
Reviewer 3 Report
Comments and Suggestions for Authors
The authors response all the reviewer's comments with modification of this manuscript.